# Uterine Carcinosarcoma—A Retrospective Cohort Analysis from a Tertiary Centre on Epidemiology, Management Approach, Outcomes and Survival Patterns

**DOI:** 10.3390/cancers17040635

**Published:** 2025-02-14

**Authors:** Sarah Louise Smyth, Katherine Ripullone, Andreas Zouridis, Christina Pappa, Geraldine Spain, Aikaterina Gkorila, Amika McCulloch, Phoebe Tupper, Farhat Bibi, Negin Sadeghi, Alisha Sattar, Shmaila Siddiki, Susan Addley, Mostafa Abdalla, Federico Ferrari, Stephen Damato, Sean Kehoe, Hooman Soleymani majd

**Affiliations:** 1Frimley Health NHS Foundation Trust, Wexham SL2 4HL, UK; 2Oxford University Hospitals NHS Foundation Trust, Oxford OX3 7LE, UK; 3Salisbury NHS Foundation Trust, Salisbury SP2 8BJ, UK; 4Bedfordshire Hospitals NHS Foundation Trust, Luton LU4 0DZ, UK; 5Buckinghamshire Healthcare NHS Trust, Aylesbury HP21 8AL, UK; 6University Hospitals of Leicester NHS Trust, Leicester LE1 5WW, UK; 7Cambridge University Hospitals NHS Foundation Trust, Cambridge CB2 0QQ, UK; 8Great Western Hospitals NHS Foundation Trust, Swindon SN3 6BB, UK; 9University Hospitals of Derby and Burton NHS Foundation Trust, Derby DE22 3NE, UK; 10Guy’s and St Thomas’ NHS Foundation Trust, London SE1 9RT, UK; 11Department of Clinical and Experimental Sciences, University of Brescia, 25136 Brescia, Italy

**Keywords:** uterine carcinosarcoma, high grade, recurrence, prognosis, survival, heterologous

## Abstract

Uterine carcinosarcoma is an aggressive cancer of the womb containing carcinomatous epithelial and sarcomatous cell types. Our research investigated the details of 77 patients treated for this cancer type at our centre across a ten-year period. We found that the 5-year overall and disease-free survival rates were 46.6% and 52.1%.Involvement of the cervix and specific cell subtypes were more likely to lead to recurrence and/or death. We also found that there was no improvement in survival/recurrence when surgical treatment included the removal of lymph nodes or when patients received chemotherapy and/or radiotherapy. Further research into this virulent cancer type is required.

## 1. Introduction

Endometrial cancer represents the 6th most common female malignancy and the most common gynaecological malignancy across the world. In 2019, there were over 435,000 new cases of endometrial cancer diagnosed globally, with this rate set only to increase due to the prevalence of risk factors such as obesity and the ageing population demographic [1]. Other risk factors for endometrial cancer include ethnicity, nulliparity, late menopause, oestrogen-only hormone replacement therapy, tamoxifen use and family history of endometrial and/or colorectal cancer [2]. Incidence increases with age, with the highest in females aged 75–79 in the UK and a significant variation in mortality is linked to socioeconomic status [3]. Worldwide, over 97,000 women died due to endometrial cancer in 2020 [4].

Important prognostic predictors of endometrial cancer include histological tumour type and spread, both of which contribute to overall staging. Tumours should therefore be classified according to the WHO Classification of Tumors, Female Genital Tumors [5,6]. One high-grade histological subtype is Uterine Carcinosarcoma (UCS). Previously known as malignant Mullerian mixed tumour, UCS is an aggressive epithelial non-endometrioid endometrial cancer [3].

Tumour cells demonstrate epithelial–mesenchymal metaplastic transition and are composed of both carcinomatous (high grade, epithelial serous, endometrioid, clear cell, mixed or undifferentiated) and sarcomatous (mesodermal) components [7,8]. Usually, the epithelial component dominates. The UCS sarcomatous component can have either a homologous (contains uterine tissue types such as leiomyosarcoma, fibrosarcoma, or endometrial stromal sarcoma) or heterologous histotype (contains extrauterine tissue types such as rhabdomyosarcoma, chondrosarcoma or osteosarcoma elements) [5,9]. Heterologous differentiation occurs in 40% of UCS, as does sarcomatous dominance and these features are more likely to occur concurrently. They are also both associated with poorer survival outcomes [10]. Further to this, the introduction of The Cancer Genome Atlas (TCGA) molecular classification has contributed additional prognostic stratification [11].

UCS is rare and represents less than 5% of all endometrial cancers and nearly 20% of non-endometrioid endometrial cancers. It has a poor prognosis, accounting for approximately 15% of deaths associated with uterine malignancy. It is diagnosed at an advanced stage more often than other endometrial cancers, with 50–60% of cases being FIGO (International Federation of Gynaecology and Obstetrics) stage III-IV. Up to 30–40% of patients present with lymphadenopathy at diagnosis, whilst about 10% also exhibit visceral metastases, especially affecting the lungs. The median overall survival is less than 2 years and the 5-year overall survival rate is less than 30% (about 50% and 20% in early and advanced stages, respectively) [10].

There remains a paucity of specific guidelines for UCS management due to its rarity [8]. For this reason, as with other endometrial cancers, UCS is considered a surgically treatable disease in patients without distant metastases [12]. The surgical staging of UCS includes total hysterectomy, bilateral salpingo-oophorectomy and infracolic omentectomy due to the high risk of microscopic omental metastases at approximately 6% [13]. Pelvic lymph node staging should also be performed and recommendations include both systematic and sentinel biopsy according to the ESGO-ESTRO-ESP guidelines [14]. Minimally invasive surgery is established as the standard treatment in early-stage endometrial cancer, with equivalent outcomes for disease-free and overall survival when compared to open hysterectomy for disease confined to the uterine corpus and with better surgical outcomes and recovery. The oncological safety of laparoscopy for stage II endometrial cancer requires further investigation [15,16]. Multimodal treatment strategies include adjuvant cytotoxic chemotherapy and radiotherapy to only modest therapeutic effect, with research suggesting that recurrence will occur in more than 50% of cases despite this [8,17]. The recent results of GOG 261 and advances in molecular profiling will no doubt provide UCS patients with improved therapeutic options in the future [18,19].

The aim of the study was to assess and present the 10-year experience of the epidemiology, management approach, outcomes and survival patterns of patients with UCS managed in a single tertiary cancer referral centre in the United Kingdom, in order to investigate independent risk factors for relapse and mortality and inform future clinical practice.

## 2. Materials and Methods

This is a retrospective cohort study using ten years of observational data from Oxford University Hospitals NHS Foundation Trust. This is a large tertiary referral cancer centre in the UK and is part of the Thames Valley Cancer Alliance Network. This study forms part of a larger body of work investigating histological types of gynaecological uterine cancers in the same cohort [20,21]. We analysed all patients who had a diagnosis of uterine carcinosarcoma on their final histology report to characterise which risk factors, comorbidities and management pathways impacted the severity of the disease encompassing diagnosis, progression and long-term survival. We included all patients with UCS treated across the Thames Valley Cancer Alliance Network (five recruiting sites, serving a patient catchment area of 2.3 million) between March 2010 and January 2020. One patient with inadequate follow-up data was excluded. Patients who did not have primary surgical treatment (*n* = 4) were excluded from our cohort as this is not considered to be standard in the management of UCS. Therefore, we analysed 77 cases of UCS. We collected all outcome measures up until October 2022 for patients.

All data was extracted retrospectively from the electronic records of patients and anonymised in the context of service evaluation for endometrial cancer and in accordance with the Oxford University Hospitals NHS Foundation Trust requirements (registration number 5832). All patients consented to data collection and analysis for approved research. This research and manuscript conform to the Helsinki Declaration, the Committee on Publication Ethics guidelines and the Reporting of studies Conducted using Observational Routinely collected health Data (RECORD) Statement validated by the Enhancing the Quality and Transparency of Health Research Network [22,23,24]. The study was not advertised. No remuneration was offered to the patients.

Patients’ demographics and comorbidities were recorded, allowing for the calculation of their Age-Adjusted Charlson Comorbidity Score (AACCS) and subsequently divided into three groups: 0–1, 2–3 and >3 [25]. Diagnostic information was gathered regarding initial biopsy histology, preoperative imaging, timings of multidisciplinary team reviews and intervals to definitive surgical management. We extracted details of treatment including surgical approach (laparotomy or laparoscopy; there were no cases of robotic laparoscopy), the performance of bilateral pelvic lymphadenectomy (and numbers of nodes yielded) and the administration of adjuvant therapy alongside final histopathological features including the depth of myometrial invasion (<50% and ≥50%); cervical stromal involvement, serosal breaching; parametrial, adnexal, pelvic and paraaortic lymph node involvement; and the presence of distant metastases, culminating in the overall FIGO stage [26]. The presence of lymphovascular invasion (LVSI) was also recorded in addition to specifics in the context of UCS regarding epithelial and sarcomatous components and overgrowth. The European Society of Gynaecological Oncology (ESGO)—European Society for Radiotherapy and Oncology (ESTRO)—European Society of Pathology (ESP) risk stratification model was used to classify all tumours [13]. Surgical staging (without uterine manipulation) and follow-up protocols were performed according to national guidelines [3] under the care of the gynaecological oncology and/or clinical/medical oncology teams. The follow-up period of data collection was from the date of surgery until October 2022 or their death. Overall survival (OS) was measured from the diagnosis date until death and disease-free survival (DFS) was measured from the date of surgery until the date of first recurrence or death from any cause.

Descriptive statistics were used to summarise the demographic and clinical characteristics of patients. Independent sample *t*-tests were used to compare continuous variables and Pearson chi-square or Fisher’s exact tests for categorical variables. Kaplan–Meier curves were used to calculate disease-free and overall survival rates and then compared using log-rank tests. To assess for potential relapse and mortality risk factors, univariate and multivariate Cox proportional hazards analyses were conducted. Statistical significance was deemed for *p* values less than 0.05 and analysis was performed using IBM©SPSS Statistics 22.0 [27].

## 3. Results

A total of 863 patients underwent surgical management of endometrial carcinoma between March 2010 and January 2020 in our tertiary centre. Uterine carcinosarcoma was diagnosed in 82 cases, representing 9.5% of all patients treated during the study period. Among them, one was excluded due to inadequate follow-up and four due to non-primary surgical management. Therefore, a total of 77 cases were analysed.

Demographic data, treatment details and clinicopathological characteristics are presented in Table 1. The mean age of the patients at diagnosis was 70 years (range 51–95) and 33% of patients had a raised Body Mass Index (BMI) of over 30 kg/m^2^. All patients were postmenopausal. The average time interval from a multidisciplinary team (MDT) discussion to treatment was 33 days and all cases underwent surgery following a standard protocol according to preoperative staging and patient performance status.

A total of 86.7% of patients were treated laparoscopically, with 80.5% (*n* = 62) undergoing pelvic lymphadenectomy, yielding an average of 13 lymph nodes (range 0–34). Pelvic lymph node histology was positive for disease invasion in 13 (21%) patients. Paraaortic lymphadenectomy was performed in 3 (3.9%) patients, 2 (67%) of whom had positive disease invasion and 52 (67.5%) patients also underwent omentectomy. The average length of hospitalisation was 3.5 days (range 1–22). The most common histopathology was represented by serous epithelial component (53.7%/36 patients) followed by high-grade endometrioid and mixed (25.4%/17 patients and 13.4%/9 patients respectively) and the majority sarcomatous component was homologous in 75.5% (*n* = 56) of cases. Lymphovascular space invasion was common in our cohort, presenting in 75.3% of cases and most cases did not feature distant metastases at 89.6% (*n* = 69).

Moreover, 58.4% of cases of UCS were treated at an early stage (stage I/II). Further, 80% (*n* = 56) of patients underwent adjuvant treatment following surgical management; 9% of patients had external beam radiotherapy only, 26% of patients had vault brachytherapy only, 35% received platinum- and taxane-based chemotherapy only, 24% received a combination of vault brachytherapy and chemotherapy and 6% received a combination of external beam radiotherapy and chemotherapy. The median follow-up after surgery was 45.56 months (range 1–138).

There were 3 cases of disease progression (defined as progression within 3 months of initial surgical treatment) and 33 (42.86%) cases of recurrence, with the average recurrence time being 20.5 months after the completion of surgical treatment (range 4–69 months). Half of these occurred within the first twelve months and 73% occurred within the first two years. In only two of our cases (6.67%) did relapse occurred more than 5 years after staging surgery. The mean recurrence to death interval was 14.53 months (range 0–118). There was no identifiable pattern of recurrence site. A total of 45 (58.4%) deaths were recorded, with 19 (42.2%) of these occurring within the first 12 months following initial treatment. The five-year overall and cancer-specific disease-free survival was 46.6% and 52.1%, respectively (Figure 1).

The univariate Cox proportional hazards analysis for the risk of recurrence and overall survival both correlated with depth of myometrial invasion (HR = 3.64, 95%CI 1.68–7.90, *p* = 0.001; and HR = 2.82, 95%CI 1.48–5.40, *p* = 0.002 respectively), cervical stromal involvement (HR = 3.68, 95%CI 1.75–7.71, *p* = 0.001; and HR = 4.77, 95%CI 2.50–9.11, *p* < 0.001 respectively), parametrial involvement (HR = 2.88, 95%CI 1.24–6.70, *p* = 0.014; and HR = 3.20, 95%CI 1.60–6.42, *p* = 0.001 respectively), an advanced disease stage (HR = 2.41, 95%CI 1.21–4.81, *p* = 0.012; and HR = 2.42, 95%CI 1.34–4.36, *p* = 0.004 respectively) and a heterologous sarcomatous component (HR = 2.20, 95%CI 1.05–4.57, *p* = 0.036; and HR = 2.08, 95%CI 1.11–3.89, *p* = 0.023 respectively). Additionally, univariate Cox proportional hazards analysis for risk of recurrence correlated with an open surgical approach (HR = 3.69, 95%CI 1.62–8.38, *p* = 0.002), adnexal involvement (HR = 2.56, 95%CI 1.11–5.92, *p* = 0.027), serosal breach (HR = 2.15, 95%CI 1.01–4.54, *p* = 0.046) and pelvic lymph node involvement (HR = 2.19, 95%CI 1.02–4.71, *p* = 0.046). Univariate Cox proportional hazards analysis for overall survival correlated with time from MDT review to surgical intervention (HR = 1.02, 95%CI 1.01–1.03, *p* = 0.026), the number of lymph nodes removed at surgery (HR = 0.96, 95%CI 0.93–0.99, *p* = 0.042), the presence of distant metastases (HR = 5.29, 95%CI 2.26–12.36, *p* < 0.001) and LVSI (HR = 2.83, 95%CI 1.20–6.70, *p* = 0.018). The univariate Cox proportional hazards analysis for the risk of recurrence and overall survival is presented in Table 2.

The multivariate Cox regression analysis after adjustment for histological, treatment and demographic characteristics including age, comorbidities (AACCS), surgical approach, pelvic lymph node dissection, depth of myometrial invasion, adnexal involvement, serosal breaching, parametrial involvement, pelvic lymph node involvement, distant metastases, LVSI and the administration of adjuvant treatment identified that cervical involvement and heterologous sarcomatous component were independently related to a higher risk of recurrence (HR = 6.269, 95%CI 1.82–21.59, *p* = 0.004; and HR = 3.62, 95%CI 1.38–9.51, *p* = 0.009 respectively), whilst only cervical involvement was independently related to a higher risk of overall mortality (HR = 3.64, 95%CI 1.42–9.38, *p* = 0.007).

Cervical stromal involvement had a significant impact on 5-year disease-free survival (27.5% vs. 60.6%, *p* < 0.001) and overall survival (14.3% vs. 59.2%, *p* < 0.001), whilst a heterologous sarcomatous component had a significant impact on 5-year disease-free survival (28% vs. 59.6%, *p* = 0.029) (Figure 2).

## 4. Discussion

Whilst this study offers a detailed analysis of one of the largest cohorts of UCS patients with long-term follow-up, it has some limitations. Our cohort included 77 patients treated over a 10-year period and followed up over 12 years. This limited our ability to detect associations for some important subcategories due to not having sufficient sample sizes. For example, the Cox regression by FIGO stage does not have sufficient numbers in each subcategory to fully elucidate any relationship between the FIGO stage and our outcomes of interest. This could be due to there being no relationship, but this would be surprising and is more likely due to a paucity of numbers. This shows the importance of collecting large cohorts of patients for even rare diseases and caution with interpretation.

The retrospective design also has some limitations. All data that we would have liked to collect was not necessarily available as we were using routine clinical data that was recorded. Furthermore, treatments offered to patients progressed over time, meaning that the groups receiving different treatments may have had different eligibility requirements and, therefore, this may have affected background risk and allowed confounding factors to confuse some of the analyses. However, this offers information on prognostics in actual clinical practice which is helpful in elucidating the centre’s data to fully inform patients.

In our cohort, the recurrence rate was 46.82%, which is higher compared to the majority of previous cohort studies (27.87–35%) [28,29,30,31]; however, we noted that this may have been due to higher numbers of early-stage disease in other cohorts. In terms of survival, our outcomes were comparable to the literature: 46.6% vs. 18–47% [10,17] 5-year overall survival and 52.1% vs. 52% [29] 5-year cancer-specific disease-free survival.

Our analysis revealed no significant difference in recurrence or mortality between those where a pelvic lymph node clearance was carried out during surgery and those where it was not (Table 1). We would expect that lymph node clearance would decrease recurrence and reduce mortality rates; however, we noted with interest that similar findings were also identified by Maiorano et al. [28,32]. This could be due to the small numbers in our study. Due to the high-grade nature of UCS, all participants would have been considered for lymph node clearance, therefore case selection should not have impacted this finding. Lymph node dissection is an invasive procedure that prolongs surgical time and complexity; therefore, if it has no impact on outcomes, we should consider whether these additional risks and costs are supported by the evidence. Part of this consideration is whether the procedure forms part of the treatment or staging (testing lymph nodes for the spread of disease).

Current clinical guidelines recommend systematic pelvic lymphadenectomy and omentectomy for UCS [10]. This is based on studies showing a survival benefit. However, current evidence is based on retrospective studies; this procedure has not been investigated through a prospective randomised controlled trial, therefore the evidence base for this recommendation requires further work and is ongoing through the Endometrial Cancer Lymphadenectomy Trial (ECLIAT) and would require further validation for UCS specifically [33].

Other surprising findings include the fact that there was no difference in our outcomes of interest comparing patients who received adjuvant therapy with those who did not, a finding that was also reported by Hapsari et al. (Table 1) [29]. According to current clinical guidelines, all UCS patients would have been offered adjuvant treatment. As with lymphadenectomy, this may not have been completed due to poor performance status, age, medical comorbidities, or patient refusal. Prior to 2019 this involved 4-6 cycles of carboplatin/paclitaxel chemotherapy followed by EBRT (extended beam radiotherapy) +/− VBT (vault brachytherapy) if the cervix was involved. After 2019 and in consideration of PORTEC 3, patients received 2 cycles of cisplatin whilst receiving EBRT followed by a further 4 cycles of carboplatin/paclitaxel +/− VBT if the cervix was involved [10]. Unlike lymph node clearance, there are a number of trials that show good improvement in recurrence and survival rates, particularly regarding the application of chemotherapy regimes, leading us to conclude this is likely due to the small sample size in our study [34,35]. Galall et al. reported in a Cochrane Systematic Review assessing 570 participants across three trials that radiotherapy did not improve survival, which is in contrast to Manzerova et al. and their extensive work in the analysis of 2342 patients with UCS using the Surveillance, Epidemiology and End Results (SEER) database [34,36].

Laparotomy (as compared to laparoscopy) was found to increase recurrence rates but had no effect on overall mortality rates. In this instance, these results are likely influenced by surgical case selection—as a more extensive disease would have been approached with laparotomy and a less severe disease with laparoscopy [37]. Therefore, we would expect higher rates of recurrence for more advanced disease as a baseline. That there was no association with increased mortality possibly points to the benefit of the more invasive procedure. Current retrospective cohorts examining UCS patients did not analyse outcomes comparing laparoscopic with open-surgical approaches for primary intervention. Nine RCTs did consider these different approaches and suggested similar disease-free survival between these two surgical techniques; however, no studies dealt with the case selection bias as above successfully, therefore the confidence in this finding is only moderate [15,38,39]. Further research with a larger sample size is required to fully elucidate the differential effect of a surgical approach on recurrence and mortality rates for UCS patients.

Cervical stromal involvement was associated with increased recurrence and mortality, as expected. This was independently related to both recurrence and overall survival after adjusting for other histological, treatment and demographic characteristics. This was also noted in our centre’s uterine group paper on high-grade endometrioid endometrial cancer [21]. The large effect size (Table 2) highlights the importance of correctly characterising cervical invasion and the importance of minimal cervical manipulation during surgical staging to avoid contributing to an adverse outcome [40].

Our study also found that there were significantly worse outcomes for those individuals with a heterologous sarcomatous component, with a similar trend identified by Perna et al. and to statistical significance by Zhu et al. (Table 2) [35,41]. Histopathological findings are central features of the 2023 revision of the FIGO staging of endometrial carcinoma based on advances in molecular classification as well as tumour patterns and histological types to better reflect our understanding of complex types of endometrial carcinoma and their biological behaviour; to improve appropriate surgical, radiation and systemic therapy recommendations as well as refine the future collection of outcome and survival data. Our findings suggest that sarcomatous component classification could be further used to direct treatment as well as cervical involvement, as discussed above [14,42].

We found recurrence rates to be highly variable between risk factors (with the exception of cervical involvement and sarcomatous component) as well as recurrence location. Therefore, current approaches to screening for recurrence need to be considered to improve overall detection rates. Currently, this involves speculum examination and screening for symptoms, such as vaginal bleeding, at regular intervals over a 5-year period. Some have argued that imaging to detect recurrence is more effective given this lack of association [21]. For those whom we have identified as having a higher risk of recurrence—with cervical involvement and/or heterologous sarcomatous components—this would likely improve detection and therefore result in earlier intervention, improving overall outcomes. Further analysis would be needed to understand if such an approach—including both clinical and imaging assessment—should be rolled out for all women and whether it would balance improved detection with increased cost.

The Cancer Genome Atlas (TCGA) has formed the foundation of a more detailed stratification of endometrial cancer from histopathological to genomic-based classifications [43]. This has introduced four molecular subtypes: (i) POLE (ultra-mutated) tumours, (ii) microsatellite unstable tumours, (iii) copy-number-high tumours with mostly TP53 mutations and (iv) the remaining group without these alterations [44]. In UCS, POLE mutations show a favourable prognosis, whereas TP53 mutations and NSMP (no specific molecular profile) indicate poor prognoses [11]. Data regarding molecular classification was limited within our cohort and therefore was not included in our analyses.

Importantly, recent studies have shown differential survival by race for UCS [45]. This can partially be attributed to socioeconomic factors leading to later diagnosis and treatment, as well as treatment regimens offered. However, analyses of the TCGA by racial group have shown that Black patients have different tumour gene mutation profiles compared with Caucasian and Asian patients in high-grade uterine cancers. Therefore, any further research to develop characterisation and subsequent treatment guidelines based on genetic and molecular markers must be sensitive to these differences to ensure equitable treatment across all minority groups.

Within our study cohort, race was poorly documented, limiting our ability to analyse our outcomes of interest and risk factors by race. A total of 47 out of 77 participants had a documented race, but of these 47, 13 were designated as ‘other’. This points to a limitation of retrospective studies in general as well as highlighting an important point that perhaps race is not being considered clinically useful for these patients. Numerous studies have highlighted the need to tailor and adjust NHS services to bridge the gap created by socioeconomic differences in a patient’s ability to engage with the healthcare service. More worrying, though, is that these differences, which are associated with racial divisions, could also form the basis of differences in carcinogenesis and treatment responses, indicating a need for it to be seriously considered clinically and in any further research of UCS.

## 5. Conclusions

We provide a detailed description of a cohort with UCS within a single UK cancer centre across the course of ten years, characterising disease, treatment approaches and subsequent clinical outcomes. Our study suggests that cervical stromal involvement and heterologous sarcomatous components are the only independent risk factors for recurrence and/or overall mortality for UCS after adjusting for other pathological and treatment parameters. More work is required to understand the role of surgical techniques (lymphadenectomy, laparotomy and cervical manipulation) as well as associated treatments (specific adjuvant regimens and biologics) on outcomes [46]. We need better evidence for the treatment approaches we are offering now as well as more work to direct what future treatments should be offered [8,10]. Movement towards molecular characterisation and immunotherapy [10] to direct treatment has already begun, but we must ensure that this includes the consideration of different groups (for example, by race [47]) to develop precision medicine in achieving the best clinical outcomes for all women.

## Figures and Tables

**Figure 1 cancers-17-00635-f001:**
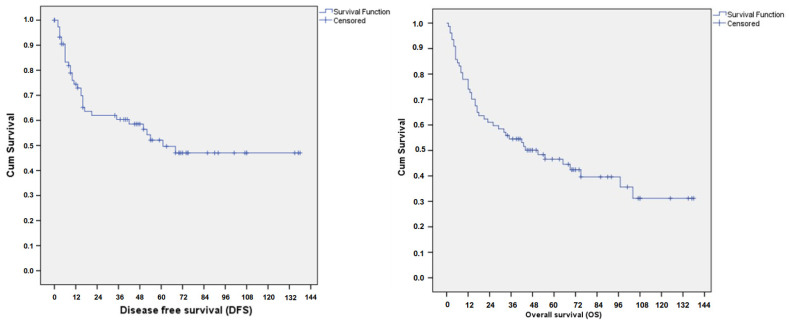
Disease-free (**left**) and overall survival (**right**) for patients with uterine carcinosarcoma.

**Figure 2 cancers-17-00635-f002:**
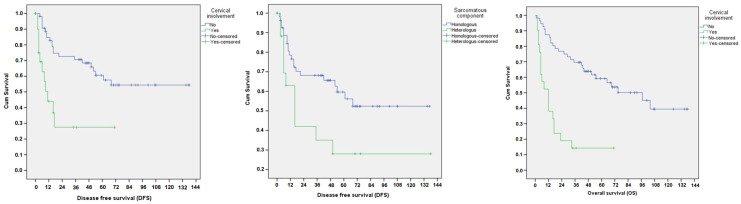
Disease-free cervical involvement (**left**), sarcomatous component (**middle**) and overall survival cervical involvement (**right**) for patients with uterine carcinosarcoma.

**Table 1 cancers-17-00635-t001:** Demographic data, treatment details and clinicopathological characteristics.

		*n* (%)	Recurrence (% of Each Subgroup)	*p*-Value	Overall Mortality (% of Each Subgroup)	*p*-Value
Demographic data	AGE			0.061		0.052
	<65	20 (26)	5 (25)		8 (40)	
	≥65	57 (74)	28 (49.1)		37 (64.9)	
	AACCIS			0.135		0.158
	0–1	5 (6.5)	0 (0)		1 (20)	
	2–3	35 (45.5)	16 (45.7)		20 (57.1) 24 (64.9)	
	>3	37 (48.1)	17 (45.9)			
Treatment details	Surgical approach			0.015 *		0.181
	Laparoscopy	65 (86.7)	24 (36.9)		36 (55.4)	
	Laparotomy	10 (13.3)	8 (80)		8 (80)	
	Pelvic lymph node (LN) dissection			0.803		0.192
	No	15 (19.5)	6 (40)		11 (73.3)	
	Yes	62 (80.5)	27 (43.5)		34 (54.8)	
	Adjuvant treatment			0.546		0.627
	No	14 (20)	5 (35.7)		9 (64.3)	
	Yes	56 (80)	25 (44.6)		32 (57.1)	
Histological features	FIGO Stage			0.150		0.106
	IA	27 (35.1)	7 (25.9)		10 (37)	
	IB	11 (14.3)	5 (45.5)		6 (54.7)	
	II	7 (9.1)	4 (57.1)		5 (71.4)	
	IIIA	10 (13)	45 (50)		6 (60)	
	IIIB	2 (2.6)	2 (100)		2 (100)	
	IIIC1	10 (13)	7 (70)		7 (70)	
	IIIC2	2 (2.6)	1 (50)		2 (100)	
	IVB	8 (10.4)	2 (25)		7 (87.5)	
	FIGO Stage category			0.125		0.013 *
	Early (I–II)	45 (58.4)	16 (35.6)		21 (46.7)	
	Advanced (III–IV)	32 (41.6)	17 (53.1)		24 (75)	
	Depth of myometrial invasion			0.017 *		0.003 *
	<50%	33 (42.9)	9 (27.3)		13 (39.4)	
	≥50%	44 (57.1)	24 (54.5)		32 (72.7)	
	Cervical stromal involvement			0.121		0.03 *
	No	56 (72.7)	21 (37.5)		27 (48.2)	
	Yes	21 (27.3)	12 (57.1)		18 (85.7)	
	Adnexal involvement			0.089		0.039 *
	No	67 (87)	26 (33.8)		36 (53.7)	
	Yes	10 (13)	7 (70)		9 (90)	
	Serosal breach			0.214		0.175
	No	59 (76.6)	23 (39)		32 (54.2)	
	Yes	18 (23.4)	10 (55.6)		13 (72.2)	
	Parametrial involvement			0.238		0.011 *
	No	65 (84.4)	26 (40)		34 (53.2)	
	Yes	12 (15.6)	7 (58.3)		11 (91.7)	
	Pelvic lymph node involvement			0.035 *		0.138
	No	64 (83.1)	24 (37.5)		35 (54.7)	
	Yes	13 (16.9)	9 (69.2)		10 (76.9)	
	Paraaortic lymph node involvement			0.836		0.508
	No	75 (97.4)	32 (42.7)		43 (57.3)	
	Yes	2 (2.6)	1 (50)		2 (100)	
	Distant metastases			0.454		0.130
	No	69 (89.6)	31 (44.9)		38 (55.1)	
	Yes	8 (10.4)	2 (25)		7 (87.5)	
	LVSI			0.252		0.006 *
	No	19 (24.7)	6 (31.6)		6 (31.6)	
	Yes	58 (75.3)	27 (46.6)		39 (67.2)	
	Sarcomatous component			0.079		0.013 *
	Homologous	56 (75.7)	21 (37.5)		28 (50)	
	Heterologous	18 (24.3)	11 (61.1)		15 (83.3)	
	Epithelial component			0.138		0.478
	Low grade endometrioid	1 (1.5)	0 (0)		0 (0)	
	High grade endometrioid	17 (25.4)	3 (17.6)		7 (41.2)	
	Serous	36 (53.7)	17 (47.2)		22 (61.1)	
	Clear cell	1 (1.5)	0 (0)		1 (100)	
	Mixed	9 (13.4)	6 (66.7)		6 (66.7)	
	Undifferentiated	3 (4.5)	1 (33.3)		0 (0)	

* For statistically significant results (*p*-value < 0.05).

**Table 2 cancers-17-00635-t002:** Univariate Cox proportional hazards analysis for the risk of recurrence and overall mortality for uterine carcinosarcoma.

	Recurrence		Death	
	HR (95% CI)	*p*-Value	HR (95% CI)	*p*-Value
AGE	1.02 (0.99–1.06)	0.204	1.02 (0.99–1.05)	0.310
AACCS	1.05 (0.88–1.25)	0.614	1.11 (0.96–1.29)	0.170
BMI	1.04 (0.97–1.12)	0.291	1.02 (0.96–1.08)	0.605
MDT to Theatre	1.01 (0.98–1.02)	0.798	1.02 (1.01–1.03)	0.026 *
Surgical approach				
Laparoscopy				
Laparotomy	3.69 (1.62–8.38)	0.002 *	1.58 (0.73–3.42)	0.247
Pelvic lymph node dissection				
No				
Yes	0.85 (0.34–2.01)	0.666	0.55 (0.28–1.11)	0.094
Number of LN removed	0.97 (0.93–1.01)	0.149	0.96 (0.93–0.99)	0.042 *
Adjuvant treatment				
No				
Yes	1.41 (0.54–3.69)	0.483	0.88 (0.42–1.86)	0.739
FIGO Stage				
IA				
IB	2.83 (0.89–8.97)	0.077	1.81 (0.66–5.00)	0.250
II	3.51 (1.02–12.13)	0.047 *	3.648 (1.22–10.89)	0.020 *
IIIA	3.00 (0.95–9.50)	0.062	2.40 (0.87–6.65)	0.091
IIIB	15.36 (3.00–78.92)	0.001 *	7.10 (1.48–34.00)	0.014 *
IIIC1	3.62 (1.27–10.34)	0.016 *	2.33 (0.88–6.15)	0.088
IIIC2	9.38 (1.08–81.33)	0.042 *	9.06 (1.91–42.98)	0.006 *
IVB	4.93 (0.98–24.79)	0.053	10.78 (3.87–30.06)	<0.001 *
Stage category				
Early (I–II)				
Advanced (III–IV)	2.41 (1.21–4.81)	0.012 *	2.42 (1.34–4.36)	0.004 *
Depth of myometrial invasion				
<50%				
≥50%	3.64 (1.68–7.90)	0.001 *	2.82 (1.48–5.40)	0.002 *
Cervical stromal involvement				
No				
Yes	3.68 (1.75–7.71)	0.001 *	4.77 (2.50–9.11)	<0.001 *
Adnexal involvement				
No				
Yes	2.56 (1.11–5.92)	0.027 *	2.02 (0.97–4.20)	0.061
Serosal breach				
No				
Yes	2.15 (1.01–4.54)	0.046 *	1.89 (0.98–3.60)	0.057
Parametrial involvement				
No				
Yes	2.88 (1.24–6.70)	0.014 *	3.20 (1.60–6.42)	0.001 *
Pelvic lymph node involvement				
No				
Yes	2.19 (1.02–4.71)	0.046 *	1.56 (0.77–3.16)	0.216
Para-aortic lymph node involvement				
No				
Yes	3.52 (0.46–26.88)	0.221	3.82 (0.91–16.08)	0.067
Distant metastases				
No				
Yes	1.96 (0.45–8.43)	0.368	5.29 (2.26–12.36)	<0.001 *
LVSI				
No				
Yes	1.94 (0.80–4.71)	0.142	2.83 (1.20–6.70)	0.018 *
Sarcomatous component				
Homologous				
Heterologous	2.20 (1.05–4.57)	0.036 *	2.08 (1.11–3.89)	0.023 *

HR—hazard ratio; * for statistically significant results (*p*-value < 0.05).

## Data Availability

Data are unavailable due to privacy restrictions.

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
