# Peer review of "Uterine Carcinosarcoma—A Retrospective Cohort Analysis from a Tertiary Centre on Epidemiology, Management Approach, Outcomes and Survival Patterns"

_cancers, 2025, doi:10.3390/cancers17040635_

Round 1

Reviewer 1 Report

Comments and Suggestions for Authors

Survival analysis is a suitable statistical analysis method for prospectively planned studies. In this study, the scope of use of KM survival analysis and Cox regression analysis has been exceeded.

In retrospective studies, especially in samples selected from the database, the characteristics that should be present in all cases may not be present.

Therefore, the research question may be missing or may not arise.

The recommended method for this data set is to plan it retrospectively as a case-control study, to use OR (Odds Ratio) instead of HR, and to use the logistic regression model instead of the Cox regression model.

If the data set meets all the requirements, the incidence rates and RR (Relative Risk) can be calculated.

Author Response

Reviewer 1 comments:

Survival analysis is a suitable statistical analysis method for prospectively planned studies. In this study, the scope of use of KM survival analysis and Cox regression analysis has been exceeded. In retrospective studies, especially in samples selected from the database, the characteristics that should be present in all cases may not be present. Therefore, the research question may be missing or may not arise. The recommended method for this data set is to plan it retrospectively as a case-control study, to use OR (Odds Ratio) instead of HR, and to use the logistic regression model instead of the Cox regression model. If the data set meets all the requirements, the incidence rates and RR (Relative Risk) can be calculated.

Reviewer 1 response:

Thank you for your feedback, we appreciate your shared knowledge of this important subject. Survival analysis with Kaplan-Meier and Cox proportional hazards regression is widely used for retrospective data in medical research. The Kaplan-Meier display of survival data has the advantage of requiring minimal assumptions while handling such data. As with any data analysis method, it is only as good as the underlying data allows. The patients of our study were treated in a single centre, by one multidisciplinary team and had the same treatment and follow up approach, therefore their management is very homogeneous, however we recognize the limitation of the retrospective nature of our data. All patients with UCS were included in our study (lines 121-130), this was not a sample selected from a database and there was no control group included for comparison, hence we felt that a cohort study would offer best representation and went on to perform analysis accordingly. Kaplan-Meier curves by themselves don't provide an efficient way to account for confounders. That's why to investigate potential risk factors for adverse prognosis we performed Cox proportional hazards regression, to try to account for potential confounders. We understand that the same dataset can be analysed in many different ways, but we feel that the selected survival analysis does not violate the statistical assumptions.

Reviewer 2 Report

Comments and Suggestions for Authors

This is a relevant and high-quality study on carcinosarcoma. The presentation of the results is clear and considers most clinically relevant aspects. The discussion is well-researched. In my opinion, the study has no significant weaknesses. I only have one minor suggestion: according to the journal's guidelines, a "simple summary" is required before the abstract.

Author Response

Reviewer 2 comments:

This is a relevant and high-quality study on carcinosarcoma. The presentation of the results is clear and considers most clinically relevant aspects. The discussion is well-researched. In my opinion, the study has no significant weaknesses. I only have one minor suggestion: according to the journal's guidelines, a "simple summary" is required before the abstract.

Reviewer 2 response:

Thank you very much for your positive feedback for which we are very grateful. As per your advice we have included a simple summary before the abstract which can be found across lines 21-27.

Reviewer 3 Report

Comments and Suggestions for Authors

1.     In lines 166-167, “The average time interval from multidisciplinary team (MDT) discussion to surgical treatment was 33 days.” The interval seems relatively longer than other hospitals. Could you explain the reason?

2.     In line 133, “surgical approach (laparotomy or laparoscopy)”, did the laparoscopy including robotic method?

3.     Is the choice of surgical method influenced by the uterine size?

4.     Did you perform modified radical hysterectomy for the stage II (Cervical stromal involvement) UCS?

5.     Did you routinely perform para-aortic lymphadenectomy?

6.     Did you use post-operation adjuvant treatment of concurrent chemo-radiotherapy or combined sequential chemotherapy with radiotherapy?

7.     Where is the recurrence site of the UCS in your study?

8.     Did the UCS uterine “tumor” size have the impact of survival?

Author Response

Reviewer 3 comments:

  1. In lines 166-167, “The average time interval from multidisciplinary team (MDT) discussion to surgical treatment was 33 days.” The interval seems relatively longer than other hospitals. Could you explain the reason?
  2. In line 133, “surgical approach (laparotomy or laparoscopy)”, did the laparoscopy including robotic method?
  3. Is the choice of surgical method influenced by the uterine size?
  4. Did you perform modified radical hysterectomy for the stage II (Cervical stromal involvement) UCS?
  5. Did you routinely perform para-aortic lymphadenectomy?
  6. Did you use post-operation adjuvant treatment of concurrent chemo-radiotherapy or combined sequential chemotherapy with radiotherapy?
  7. Where is the recurrence site of the UCS in your study?
  8. Did the UCS uterine “tumor” size have the impact of survival?

Reviewer 3 response:

  1. Thank you for highlighting this important point. Whilst all UCS patients were treated in a single tertiary centre, they were referred from 4 secondary care cancer units initially. Owing to variations in administrative activities, transfer of information and appointment arrangements alongside the constant significant workload of the cancer centre team, we deem that this likely had an impact on the MDT to surgical treatment interval. We have not chosen to include this as a major point within our discussion owing to the other points which we felt warranted greater merit.
  2. This did not include robotic method. We have edited the manuscript accordingly to reflect this in lines 148-149.

3&4. The choice of surgical method including type of hysterectomy and approach was influenced by preoperative stage of disease as discussed in lines 317-330 and in consideration of patient fitness for treatment, as per standard practice for management of UCS. We have updated our manuscript to reflect this in lines 187-188.

  1. No we did not routinely perform paraaortic lymphadenectomy as outlined in lines 193-194. We have not included paraaortic lymphadenectomy when discussing UCS surgical staging in lines 94-98.
  2. Adjuvant treatment was administered according to patient choice and discussion in consideration of risk/benefit and performance status/patient fitness. Prior to 2019 this involved 4-6 cycles of carbo/taxol chemotherapy followed by EBRT +/- VBT if the cervix was involved. After 2019 and in consideration of PORTEC 3, patients received 2 cycles of cisplatin whilst receiving EBRT followed by a further 4 cycles of carbo/taxol +/- VBT if the cervix was involved.
  3. As documented on line 215 there was no identifiable pattern of recurrence site.
  4. With regret we did not include tumour size in our analysis when assessing foctors which had an impact on survival and we will also consider this for future research into this important topic. We are grateful that this has been brought to our research groups attention going forward.

Round 2

Reviewer 1 Report

Comments and Suggestions for Authors

The explanations made by the author are reasonable and acceptable.